# Occurrence of *Brucella ceti* in striped dolphins from Italian Seas

Giuliano Garofolo[1], Antonio Petrella[2], Giuseppe Lucifora[3], Gabriella Di Francesco[1], Giovanni Di Guardo[4], Alessandra Pautasso[5], Barbara Iulini[6], Katia Varello[6], Federica Giorda[6,7], Maria Goria[6], Alessandro Dondo[6], Simona Zoppi[6], Cristina Esmeralda Di Francesco[4], Stefania Giglio[8], Furio Ferringo[2], Luigina Serrecchia[2], Mattia Anna Rita Ferrantino[2], Katiuscia Zilli[1], Anna Janowicz[1], Manuela Tittarelli[1], Walter Mignone[6], Cristina Casalone[6], Carla Grattarola[6]*

1 National and OIE Reference Laboratory for Brucellosis, Istituto Zooprofilattico Sperimentale dell'Abruzzo e del Molise, Teramo, Italy, 2 Istituto Zooprofilattico Sperimentale della Puglia e della Basilicata, Foggia, Italy, 3 Istituto Zooprofilattico Sperimentale del Mezzogiorno, Vibo Valentia, Italy, 4 Faculty of Veterinary Medicine, University of Teramo, Teramo, Italy, 5 ASL.1 Imperiese, Imperia, Italia, 6 OIE Collaborating Centre Health of Marine Mammals, Istituto Zooprofilattico Sperimentale del Piemonte, Liguria e Valle d'Aosta, Torino, Italy, 7 Institute for Animal Health and Food Safety (IUSA), Faculty of Veterinary Medicine, University of Las Palmas de Gran Canaria, Las Palmas de Gran Canaria, Canary Islands, Spain, 8 M.A.R.E. Calabria Association, Montepaone (Catanzaro), Italy

* carla.grattarola@izsto.it

**Data Availability Statement:** The WGS data is submitted at NCBI GenBank with the following accession number: PRJNA623338.

## Abstract

*Brucella ceti* infections have been increasingly reported in cetaceans, although a very limited characterization of Mediterranean *Brucella* spp. isolates has been previously reported and relatively few data exist about brucellosis among cetaceans in Italy. To address this gap, we studied 8 cases of *B. ceti* infection in striped dolphins (*Stenella coeruleoalba*) stranded along the Italian coastline from 2012 to 2018, investigated thanks to the Italian surveillance activity on stranded cetaceans. We focused on cases of stranding in eastern and western Italian seas, occurred along the Apulia (N = 6), Liguria (N = 1) and Calabria (N = 1) coastlines, through the analysis of gross and microscopic findings, the results of microbiological, biomolecular and serological investigations, as well as the detection of other relevant pathogens. The comparative genomic analysis used whole genome sequences of *B. ceti* from Italy paired with the publicly available complete genomes. Pathological changes consistent with *B. ceti* infection were detected in the central nervous system of 7 animals, showing non-suppurative meningoencephalitis. In 4 cases severe coinfections were detected, mostly involving *Dolphin Morbillivirus* (DMV). The severity of *B. ceti*-associated lesions supports the role of this microbial agent as a primary neurotropic pathogen for striped dolphins. We classified the 8 isolates into the common sequence type 26 (ST-26). Whole genome SNP analysis showed that the strains from Italy clustered into two genetically distinct clades. The first clade comprised exclusively the isolates from Ionian and Adriatic Seas, while the second one included the strain from the Ligurian Sea and those from the Catalonian coast. Plotting these clades onto the geographic map suggests a link between their phylogeny and topographical distribution. These results represent the first extensive characterization of *B. ceti* isolated from Italian

**Funding:** This work was supported by the Italian Ministry of Health. WM received funding from the Italian Ministry of Health under gran agreement code IZSPLV 09/18. GG received funding from the Italian Ministry of Health under gran agreement code IZSAM 02/17 RC. The funder had no role in study design, date collection and analysis, decision to publish, or preparation of the manuscript. There was no additional external funding received for this study.

**Competing interests:** The authors have declared that no competing interests exist.

waters reported to date and show the usefulness of WGS for understanding of the evolution of this emerging pathogen.

## Introduction

*Brucella* is a genus of bacteria that infects many terrestrial and aquatic vertebrates [1] and brucellosis represents a widespread zoonosis and an important economic and public health problem in many areas of the world [2].

*Brucella* spp. infections were first described in pinnipeds and cetaceans in the early 1990's [3, 4], in California and Scotland, and have been reported since then in several wild marine mammal species all over the world. Since 2007, isolates of *Brucella* spp. from marine mammals have been classified further based on molecular investigations, differences in metabolism and host-bacteria interactions, into two species, *B. ceti* and *B. pinnipedialis* that infect cetaceans and pinnipeds, respectively [5].

The occurrence of a variety of lesions caused by brucellosis in cetaceans, such as endometritis, placentitis, abortion, orchitis, mastitis, pneumonia, myocarditis, pericarditis, osteoarthritis, spinal discospondylitis, subcutaneous abscesses, hepatic, splenic or lymph node necrosis, macrophage infiltration in liver and spleen, and meningoencephalomyelitis has been demonstrated [6–18]. However, compared with the reported high global seroprevalence of marine mammal *Brucella* spp. infection, clinical disease does not appear to be common [7, 11, 19, 20], suggesting that most of the infected animals overcome clinical disease, eventually remaining *Brucella* carriers and shedders [14].

Based on the common infection patterns of meningitis and/or meningoencephalitis, resembling "human neurobrucellosis" [21, 22], a specific susceptibility has been suggested for the striped dolphin (*Stenella coeruleoalba*) [9, 11, 19, 23–26]. Central nervous system (CNS) involvement in *Brucella* infection in man occurs in about 5–7% of the cases [1], and includes findings of meningitis, encephalitis, meningovascular disease, brain abscesses and demyelinating syndrome [21].

The dynamics of *B. ceti* infection in cetaceans are largely unknown, and the cell receptor(s) allowing the entry of this pathogen and the subsequent dissemination throughout cetacean host's tissues have not yet been identified [27]. Although the role of metazoan parasites in the eco-epidemiology and pathogenesis of brucellosis in cetaceans is unclear, the localization of *B. ceti* in lungworms and cestoda raise the possibility that they may serve as carriers for the transmission of the infection [11, 13].

According to Multi Locus VNTR (Variable copy of Tandem Repeats) Analysis (MLVA), the marine *Brucella* strains can be divided into three major groups containing eight clusters [28, 29], and the Multi Locus Sequence Typing (MLST) identified 15 sequence types (STs) so far as reported at the public PubMLST repository (https://pubmlst.org/brucella/). Marine mammal *Brucella* strains have potential to infect and cause disease in domestic animals [30] and humans. However, only few human clinical cases have been observed to date and linked mainly to raw fish and seafood ingestion [31, 32] or to laboratory operations without proper biosafety containment [33–35].

*B. ceti* infections have been frequently described in dolphins from both, the Atlantic and Pacific Oceans, but to date the information about isolates from marine mammals of Mediterranean Sea is limited and relatively little data exist about brucellosis in cetaceans in Italy. The detection of anti-*Brucella* spp. antibodies was first demonstrated in cetaceans stranded along

the Spanish coast of the Mediterranean from 1997 to 1999 [36], while no evidence of seropositivity was detected in cetaceans of Italian Seas until the beginning of 2015 [16, 37].

The isolation of *B. ceti* in the Mediterranean Sea was first achieved in 2009, in a striped dolphin stranded along the Spanish Catalonian coast [14]. Later, in 2012, the presence was documented in four other cetaceans, two of them, one striped dolphin and one bottlenose dolphin, stranded in Spain along the same Catalonian coastline [14], and the other two, both striped dolphins, stranded in Italy, in the Tyrrhenian and the Adriatic Sea [26, 38]. Based on MLST investigations all *B. ceti* strains isolated in these cases belonged to ST-26 [38].

Recently, a coinfection by *Brucella* spp., *Listeria monocytogenes* and *Toxoplasma gondii* was confirmed using molecular methods in a striped dolphin stranded in Italy along the Ligurian coastline, and was associated with related pathological changes in brain, blubber, liver and spleen [16]. Moreover, a *B. ceti* ST-27 strain, previously found only in Pacific Ocean species [11, 39], was isolated from multiple lymph nodes of one bottlenose dolphin in the Croatian part of the northern Adriatic Sea, representing the evidence of this zoonotic strain in Mediterranean waters [40, 41] and in European waters in general.

In order to gain a deeper understanding about brucellosis in cetaceans in Italy, we studied eight cases of *B. ceti* infection detected in striped dolphins (*S. coeruleoalba*) stranded along the Italian shoreline from 2012 to 2018. We focused on the pathogenic role shown by the pathogen and on the genetic constitution of the strains involved, subjected to a comparative genomic analysis in order to determine the relationship of Italian isolates to those previously described in European/Mediterranean waters [42].

## Materials and methods

### Ethics statement

The National Reference Centre for Diagnostic Investigations on Stranded Marine Mammals (C.Re.Di.Ma.–Istituto Zooprofilattico Sperimentale del Piemonte, Liguria e Valle d'Aosta, Torino, Italy) and the Istituti Zooprofilattici Sperimentali (IIZZSS) are public laboratories authorized by the Italian Ministry of Health to perform systematic surveys on infectious diseases of aquatic mammals stranded on the cost of Italy. This study was done through passive surveillance sampling animals found dead, therefore the procedures applied did not harm live animals.

### Dolphins

We investigated 8 cases of stranding, associated with striped dolphins found stranded dead along the Apulian (Adriatic and Ionian Seas), Ligurian (Ligurian Sea) and Calabrian (Ionian Sea) coastlines, from 2012 to 2018 that resulted positive to the isolation of *B. ceti* from one or more tissues.

These positive cetaceans were distributed along the Apulian (N = 6), Calabrian (N = 1) and Ligurian (N = 1) coastlines (Fig 1).

### Necropsy and diagnostic investigations

At the time of stranding, the carcasses were submitted to the diagnostic laboratories, belonging to the network of Istituti Zooprofilattici Sperimentali laboratories, coordinated by the C.Re.Di. Ma, where a detailed post-mortem examination was performed according to standard protocols [43], depending on the carcasses' preservation status.

Individual data, including date and location, sex, age class categories (based on total body length-TBL), decomposition code [43] and body condition [44] were recorded. The presence

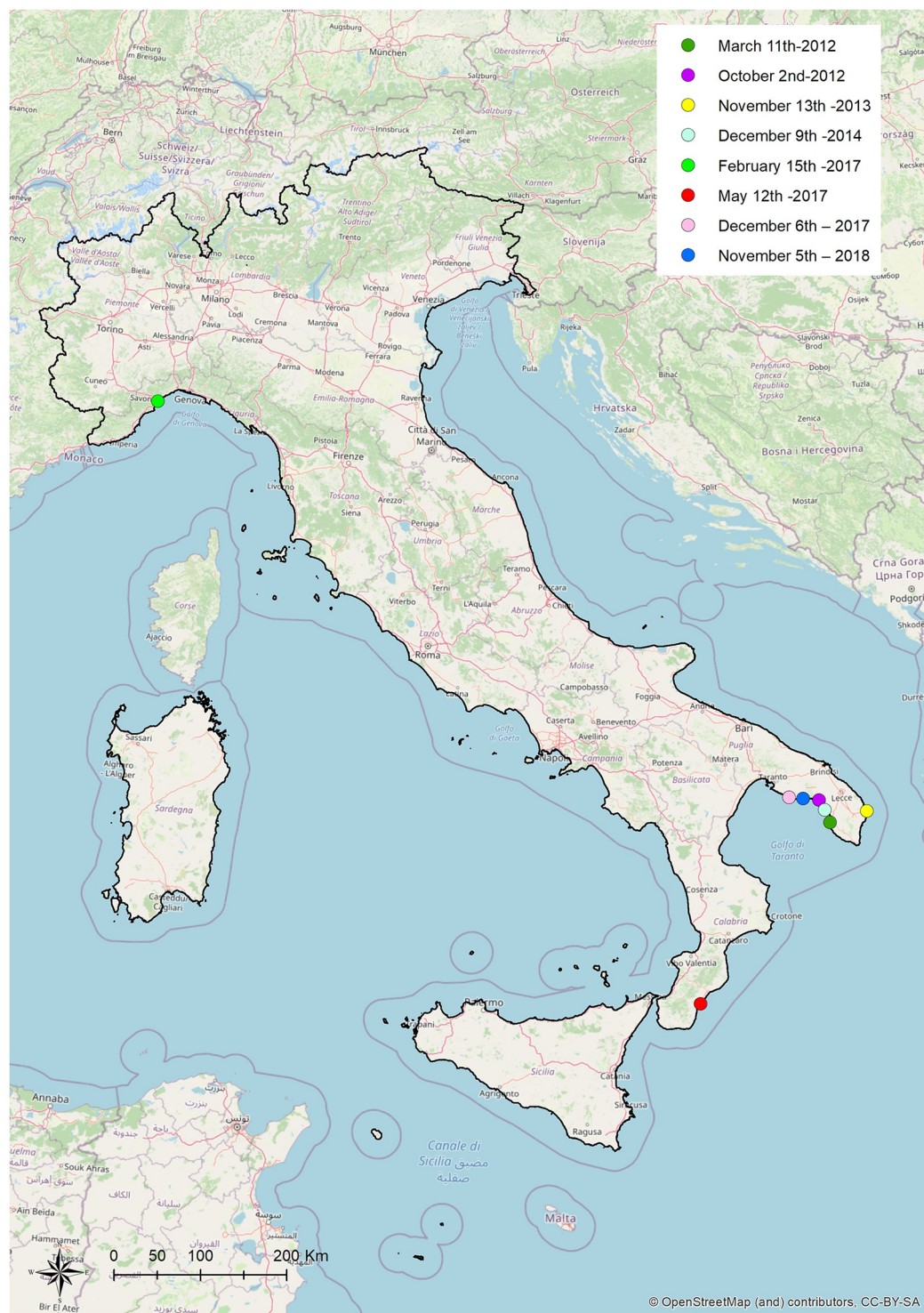

**Fig 1. Stranding sites of striped dolphins infected by *B. ceti* under study, Italy, 2012–2018.** Geographical mapping was obtained by ArcGIS® software using the geographical coordinates found from the strandings. Location data (dots): Case 1 dark green: Gallipoli Lido Pizzo, Apulian coastline, Ionian Sea (39.58 N, 18.00 E) March 11th 2012. Case 2 purple: Porto Cesareo Bacino Grande, Apulian coastline, Ionian Sea (coordinates not available) October 2nd 2012. Case 3 yellow: Alimini, Otranto, Apulian coastline, Adriatic Sea (coordinates not available) November 13th 2013. Case 4 light blue: Porto Cesareo Bacino Grande, Apulian coastline, Ionian Sea (coordinates not available) December 9th 2014. Case 5 green: Savona, on the Ligurian coastline, Ligurian sea (44.94 N, 8.18 E) February 15th 2017. Case 6 red: Ardore, on the Calabrian coastline, Ionian Sea (38.11 N, 16.12 E) May 12th 2017. Case 7 pink: Maruggio, on the Apulian coastline, Ionian Sea (40.17

N, 17.35 E) December 6th 2017. Case 8 blu: Manduria, on the Apulian coastline, Ionian Sea (40.18 N, 17.40 E) November 5th 2018. The Italy map was used under a CC BY-SA copyright from OpenStreetMap contributors (https://www. openstreetmap.org/copyright/en) (For interpretation of the references to colour in this figure legend, the reader is referred to the web version of this article).

of helminths was estimated by macroscopic and microscopic examination of tissues. Endoparasites were preserved in 70% alcohol for microscopic identification, according to established morphological characteristics [45, 46]. Coronal sections from different brain regions (telencephalon, diencephalon, mesencephalon, pons, cerebellum, medulla) [47], as well as from all major organs, were fixed in 10% neutral buffered formalin, embedded in paraffin, sectioned at 4 μm, stained with hematoxylin and eosin, and finally examined under a light microscope.

The presence of relevant pathogens like DMV and *T. gondii* was investigated by PCR methods as previously reported [48, 49]. Frozen CNS tissue samples of all animals and additional organs for cases 5, 6 and 7 were tested for both pathogens.

In one case (case 5), the presence of DMV and *T. gondii* was also explored through immunohistochemical investigation (IHC) [50], on CNS and bladder sections, for DMV, and CNS sections, for *T. gondii*, respectively.

Serological investigations to estimate the presence of specific antibodies against DMV and *T. gondii* were also performed in two cases (5, 6) [50], specifically on serum, cerebrospinal fluid (CSF) and aqueous humor of case 5, and on serum of case 6.

Macroscopical and microscopical findings of all cases, except case 3, were recorded.

Selected tissues and/or fluids were collected for microbiological, biomolecular and serological investigations focused on *Brucella* infection diagnosis.

## *Brucella* isolation and identification

The primary isolation of *Brucella* spp. was performed from CNS samples of all animals, and from other tissues available for cases 1, 5, 6, 7 and 8.

Specifically, the isolation of *Brucella* spp. was also attempted from spleen of Case 1, spleen, lung, prescapular lymph node and cerebrospinal fluid (CSF) of case 5, spleen, lung, lymph nodes and liver of case 6, spleen, lung, liver, kidney and testes of Case 7, and spleen, lung, mesenteric lymph nodes, liver and kidney of case 8.

The CNS of case 6 was submitted for bacterial isolation subsequently to the observation of microscopic lesions suggestive of neurobrucellosis, thanks to the histopathological analysis performed retrospectively on CNS tissue of the animal under study.

Cultures were performed according to the technique described in the OIE Manual of Diagnostic Tests and Vaccines [51], using both selective and non-selective solid media and enrichment broths to enhance the chance of isolating, except for tissues other than CNS of case 6.

For cases 1, 2, 3, 4, we used Farrell's and Columbia blood Agar media, for cases 7 and 8 we added modified Thayer Martin and CITA media, while for cases 5 and 6 we used a combination of Farrell's and CITA media. The solid media were incubated at 37°C, aerobically and in a microaerophilic atmosphere containing 5% $CO_2$, for at least 10 days. Enrichment cultures were carried out in *Brucella* enrichment broth, supplemented with fetal horse serum and modified *Brucella* selective supplement, and incubated at 37°C in a microaerophilic atmosphere containing 5% $CO_2$, while for cases 7 and 8 we added Thayer Martin broth. Weekly for six times or up until isolation, a loopful of the enrichment broth was streaked to Farrell's Agar medium. Suspect colonies (circular, convex, shiny, 1–2 mm in diameter after 48–72 h) were seeded onto blood agar medium and incubated for a further 2 days before re-examination. When *Brucella* spp. was suspected based on the Gram's staining [52], the colonies were tested

for catalase, oxidase and urease activities [52]. Motility and slide agglutination tests with *Brucella* anti-A and anti-M antisera were also performed for cases 1, 2, 3, 4, 7, 8, together with nitrate reduction, $H_2S$ production and growth in the presence of $CO_2$ for cases 5 and 6 [52].

For DNA extraction, all *B. ceti* isolates were subcultured in Brucella medium base (BAB; Oxoid, Hampshire, UK) and incubated in a 5–10% $CO_2$ atmosphere at 37˚C for 48 h to assess the purity of cultures and the absence of dissociation. Bacterial DNA was extracted from single colonies using Maxwell® 16 Tissue DNA Purification Kit using Maxwell® 16 Instrument (Promega, Madison, WI, USA) or High Pure DNA Template Preparation kit (Roche Diagnostics, France) according to the manufactures' instructions.

All strains isolated from the striped dolphins under study were identified as *B. ceti* using the PCR-RFLP method [53] and then subjected to genomic analysis at the National and OIE Reference Laboratory for Brucellosis, Istituto Zooprofilattico Sperimentale dell'Abruzzo e del Molise, Teramo, Italy.

### Molecular detection of *Brucella* spp. from tissues

Frozen tissue samples of cases 5, 6 and 7 were submitted to PCR for the detection of *Brucella* spp. by hemi-nested PCR targeting an outer membrane protein gene of *B. abortus* [54].

The reactions were loaded as previously reported using *B. suis* bv 2 strain Thomsen as positive control and no template control as negative control.

Specifically, molecular detection was attempted on CNS, spleen, lung, liver, pre-scapular, tracheobronchial, pulmonary lymph nodes, tongue and skin ulcers, laryngeal tonsil and CSF of case 5; on CNS of case 6; on CNS, spleen, liver, lung and testes of case 7. For DNA extraction, tissue samples (30–50 mg) were physically disrupted using a TissueLyser II homogenizer (Qiagen, Hilden, Germany) by high-speed shaking in plastic tubes with stainless steel beads (5 mm in diameter). Genomic DNA was then extracted from the disrupted tissues with an All-Prep DNA/RNA Mini kit (Qiagen) according to the manufacturer's instructions.

The PCR products were analyzed by electrophoresis on 2% agarose gel containing GelRed (Biotium, Fremont, California, USA), compared with molecular weight markers and subsequently photographed on a Gel-Doc UV transilluminator system (Bio-Rad, Hercules, California, USA).

### Serological tests for brucellosis

Serum, cerebrospinal fluid and aqueous humor from case 5, and serum from cases 6 and 8, were tested by rapid serum agglutination (Rose Bengal plate test, RBT) using RBT antigen produced from *B. abortus* strain S99 [24, 50] to detect anti-smooth *Brucella* spp. antibodies. For case 8, the test was performed on a fresh sample, while for cases 6 and 8 was carried on thawed sera.

### Whole genome sequencing and bioinformatics

Total genomic DNA of eight samples, from the cases studied with the following ID numbers 31957, 2780, 3838, 17753, 1259 and 25153, was sequenced using the Illumina NextSeq 500 platform. Briefly, the quantity of total DNA was measured with the Qubit fluorometer (QubitTM DNA HS assay; Life Technologies, Thermo Fisher Scientific, Inc.). The libraries were prepared using the Nextera XT library preparation kit (Illumina Inc., San Diego, CA) following the manufacturer's instructions and the libraries were sequenced using NextSeq 500/550 Mid Output Reagent Cartridge v2 with 300 cycles generating 150 bp paired-end reads. Reads shorter than 70 bp and average Phred mean quality < 24 were automatically discarded. Raw reads were quality assessed using FastQC and trimmed to remove nucleotides with quality score less

than 20 from 5′ end and 3′ end. Read coverage ranged from 29X to 113X, with an average of 63X.

Genomes were assembled using SPAdes version 3.11.1. and the scaffolds were used to assign MLST profiles using mlst tool (https://github.com/tseemann/mlst) which incorporates components of PubMLST database (https://pubmlst.org).

Sequence reads were deposited in Sequence Read Archive (SRA) database under NCBI Bioproject PRJNA623338.

Two samples, 10759 and 28753, were sequenced using IonTorrent platform and full genomes were assembled with Velvet 1.1.0 and deposited in NCBI with RefSeq Accession Numbers GCF_000590795.1 and GCF_000590815.1, respectively [42]. Additional 51 *B. ceti* and *B. pinnipedialis* WGS sequences available from the public database GenBank or Sequence Read Archive (SRA) were also included in the analysis. The dataset was limited to non-identical sequences that mapped to the *B. ceti* reference genome with less than 500 ambiguous matches (GenBank Accession Numbers NC_022905.1; NC_022906.1). SNP analysis was performed using In Silico Genotyper (ISG) version 0.16.10–3[55] using BWA-MEM (version 0.7.12-r1039) [56] as the aligner and GATK (version 3.9) [57] as a SNP caller. *B. ceti* genome (GenBank Accession Numbers NC_022905.1; NC_022906.1) was used as a reference. Default filters were applied to remove SNPs from duplicated regions, from regions with read coverage of less than 10X and with base call proportion less than 90%. Concatenated unique variants were used to generate maximum likelihood tree using IQ-TREE (version 1.6.9) [58, 59]. Ascertainment bias correction option was used to correct the branch lengths for the absence of constant sites in the SNP alignment. ModelFinder was used to select the best fit model and based on Bayesian Information Criterion (BIC) value and TVM+F+R2 model was chosen for phylogenetic reconstruction. Branch support was assessed using non-parametric bootstrap with 500 bootstrap replicates.

The population structure was assessed using a Bayesian approach implemented in BAPS 6.0 software with the module hierarchical BAPS (hierBAPS), which examines the existence of subgroups within a population and predicts the placement of individual sequences into specific clusters [60–62].

## Results

Post-mortem and histopathological investigations were performed on seven of the eight animals with positive culture for *B. ceti* (cases 1, 2, 4, 5, 6, 7, 8). The main gross findings included a moderate parasitic infection by *Phyllobothrium delphini* and/or *Monorygma grimaldi* plerocercoids (5/7; 71,4%), lymphadenomegaly (3/7; 42,8%), parasitic bronchopneumonia (3/7; 42,8%) and meningeal hyperaemia (3/7; 42,8%), which is regarded as the only macroscopic *Brucella* spp.-related lesion (cases 1, 4, 8) (Table 1).

The main microscopic findings were: non-suppurative meningoencephalitis (5/7; 71,4%); non-suppurative meningitis/leptomeningitis (2/7; 28,5%); multicentric lymphoid reactive hyperplasia (2/7; 28,5%); bronchointerstitial pneumonia (2/7; 28,5%); hepatocyte vacuolar degeneration (2/7; 28,5%); parasitic gastritis (2/7; 28,5%). Non-suppurative meningitis and/or meningoencephalitis, involving a mononuclear cell infiltration, observed in cases 1, 2, 4, 6, 7, 8, along with lymphoid necrosis and necrotizing hepatitis in case 5, were considered *Brucella*-type lesions (Table 1).

Moreover, protozoan cysts were apparent in the brain tissue of case 5 (S1 Table).

The microscopical features of *B. ceti*-associated CNS lesions are shown in Fig 2.

*B. ceti* was isolated from CNS of all the studied dolphins, while spleen was positive only in cases 1 and 5, and lung in case 5. The PCR for *Brucella* spp. detection directly from tissues

**Table 1. *Brucella ceti*-infected striped dolphins: stranding data, body condition, most significant findings (gross and microscopic), bacteriological, molecular and serological *Brucella* spp. investigations results, along with coinfections and the most probable cause of death.**

| Case no. | ID strain | Place, coordinates, date and stranding condition | DC | Sex | Age | NS | Main pathological findings (macro/micro) ** | *B. ceti* isolation*** | Molecular detection of *Brucella* spp *** | RBT | Coinfections | Most probable cause of death |
|---|---|---|---|---|---|---|---|---|---|---|---|---|
| 1 | 10759 | Gallipoli, (39.58 N, 18.00 E) March 11th-2012 found dead | 2 | Ma | Ju | Mo | moderate parasitic infection by *Phyllobothrium delphini* and *Monorygma grimaldi* plerocercoids; **meningeal hyperaemia**; cerebral edema; pulmonary edema; **non-suppurative leptomeningitis** | **CNS-spleen** | NP | NP | | severe cerebral impairment, associated to a primary *B. ceti* infection. |
| 2 | 28753 | Porto Cesareo*, October 2nd-2012 found dead | 3 | Fe | Ju | Go | hepatocyte vacuolar degeneration; generalized lymphoid depletion; **non-suppurative meningitis** | **CNS** | NP | NP | | severe cerebral impairment, associated to a primary *B. ceti* infection. |
| 3 | 31957 | Otranto*, November 13th -2013 found dead | ND | ND | ND | ND | data not available | **CNS** | NP | NP | | ND |
| 4 | 2780 | Porto Cesareo*, December 9th -2014 found dead | ND | Ma | Ju | Mo | **meningeal hyperemia**; cerebral edema; marked lymphomonocytic meningitis (+++ medulla oblongata), lymphomonocytic plexocoroiditis, perivascular mononuclear cuffing; **non-suppurative meningoencephalitis** | **CNS** | NP | NP | | severe cerebral impairment by a *B. ceti* infection |
| 5 | 3838 | Savona, 44.94 N, 8.18 E), February 15th -2017 found dead | 2 | Ma | Ju | Po | moderate parasitic infection by *Phyllobothrium delphini* and *Monorygma grimaldi* plerocercoids; ulcerative glossitis; skin ulcers; splenic and prescapular lymphadenomegaly; bronchointerstitial pneumonia; **multicentric lymphoid necrosis** (spleen, PSC and PUL lymph nodes, laryngeal tonsil); **multifocal necrotizing hepatitis**; interstitial nephritis; **non-suppurative meningoencephalitis** | **CNS-spleen-lung**- PSC ln-CSF | **CNS-spleen-lung, liver, PSC-TB-PUL ln** tongue and skin ulcers- laryngeal tonsil-CSF | Neg (CSF, AH, S) | DMV *T. gondii* | severe cerebral impairment, associated to a coinfection (DMV, *T. gondii*, *B. ceti*) |
| 6 | 17753 | Ardore, (38.11 N, 16.12 E) May 12th -2017 found dead | 2 | Fe | Ad | Go | moderate parasitic infection by *Phyllobothrium delphini* and *Monorygma grimaldi* plerocercoids; severe parasitic bronchopneumonia; parasitic gastritis; **non-suppurative meningoencephalitis** | **CNS**, spleen, LNs, liver, lung | **CNS** | Neg (S) | DMV | severe cerebral impairment, associated to a coinfection (DMV, *B. ceti*) |

*(Continued)*

**Table 1.** (Continued)

| Case no. | ID strain | Place, coordinates, date and stranding condition | DC | Sex | Age | NS | Main pathological findings (macro/micro) ** | *B. ceti* isolation*** | Molecular detection of *Brucella* spp *** | RBT | Coinfections | Most probable cause of death |
|---|---|---|---|---|---|---|---|---|---|---|---|---|
| 7 | 1259 | Maruggio, (40.17 N, 17.35 E) December 6th– 2017 found dead | 2 | Ma | Ju | Po | moderate parasitization by *Phyllobothrium delphini* and *Monorygma grimaldi* plerocercoids; moderate parasitic bronchopneumonia; multicentric reactive lymphadenopathy; moderate hepatocyte vacuolar degeneration; moderate multifocal bronchointerstitial pneumonia; mild multifocal mononuclear myocardial infiltrate; cerebral haemorrhages; **severe non-suppurative meningoencephalitis** | **CNS**-spleen-liver-lung-kidney-testes | CNS, spleen, liver, lung, kidney, testes | NP | DMV | severe cerebral impairment, associated to a coinfection (DMV, *B. ceti*) |
| 8 | 25153 | Manduria, (40.18 N, 17.0 E) November 5th– 2018 found dead | 2 | Ma | Ju | Mo | moderate parasitization by *Phyllobothrium delphini* plerocercoids; hemothorax and hemoperitoneum; generalized congestion; multicentric reactive lymphadenopathy; epicardial petechiae; mild parasitic bronchopneumonia; hemorrhagic parasitic gastritis; severe necrotizing enteritis; **meningeal hyperaemia**; **non-suppurative meningoencephalitis** | **CNS**-lung-liver-kidney-spleen-MES ln | NP | Pos (S) | DMV | severe cerebral impairment, associated to a coinfection (DMV, *B. ceti*) |

*: coordinates not available. DC, decomposition code (2, fresh; 3, moderate autolysis); Ma, male; Fe, female; Ad: adult; Ju: juvenile; NS, nutritional status; Mo, moderate; Po, poor; ND, not determined; NP, not performed; Neg, negative; Pos, positive; CNS: central nervous system; PSC ln, prescapular lymph node; TB ln, tracheo-bronchial lymph node; PUL ln, pulmonary lymph node; MES ln, mesenteric lymph node; LNs, lymph nodes; CSF, cerebrospinal fluid; AH, aqueous humor; S, serum; RBT, rosa bengala test; DMV, *Dolphin Morbillivirus*.

**Brucella*-associated pathological features are shown in bold.

***Tissues positive are shown in bold.

resulted positive from CNS in case 5 and 6, in two out the three animals investigated, and from six different body sites in only one case (Table 1). On the contrary, the case 7 showed negative results from all the tissues tested.

The RBT test did not detect anti-*Brucella* antibodies in samples from cases 5 and 6, while resulted positive in the serum of the case 8 (Table 1).

Cases 6, 7 and 8 showed the coinfection by *B. ceti* and DMV and case 5 demonstrated the coinfection by *B. ceti*, DMV and *T. gondii*. Specifically, the animal showed anti-*T.gondii* and anti-DVM antibodies along with positive IHC and PCR tests from different tissues.

A full description of gross and microscopic findings observed, along with complete analytical data and considerations about the cause of death and the pathogenic role of *B. ceti* infection for each case considered are available in the S1 Table.

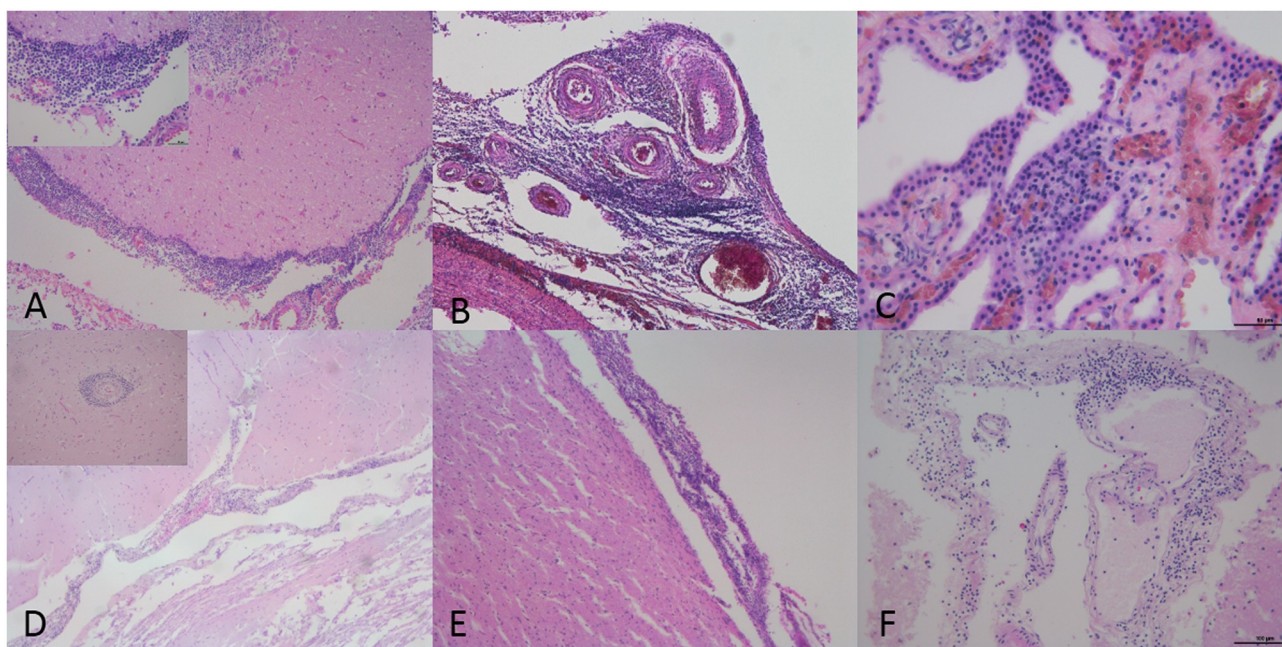

**Fig 2. *B. ceti*-associated lesions in central nervous system of striped dolphins (*S. coeruleoalba*).** (A) Severe non-suppurative meningitis. Cerebellar meninges are infiltrated by mononuclear cells (case 2). 10x. H&E. Left upper inset: detail of the lympho-monocytic inflammatory infiltrate. 40x. H&E. (B) Severe non-suppurative meningitis (case 4). Meninges at the level of medulla oblongata are infiltrated by lympho-monocytic cells. 10x. H&E. (C) Mild non-suppurative meningoencephalitis (case 5). Chorioid plexuses are infiltrated by lympho-monocytic cells. 40x. H&E. (D) Mild non-suppurative meningoencephalitis (case 6). cerebellar meninges are infiltrated by mononuclear cells. 10x. H&E. Left upper inset: perivascular cuff characterized by the presence of lympho-monocytic cells. 20x. H&E. (E) Non-suppurative meningitis (case 7). Cerebral cortex meninges are infiltrated by mononuclear cells. 10x. H&E. (F) Non-suppurative meningitis (case 8). Meninges at the level of parietal cortex are infiltrated by mononuclear cells. 20x. H&E.

The genotyping of *B. ceti* using MLST classified the 8 strains as sequence type 26 (ST-26). The publicly available *B. ceti* and *B. pinnipedialis* genomes were assigned to the STs 23, 24, 25, 26 and 27.

The SNP analysis revealed 6,320 putative polymorphisms. The mapping of sequences to the reference ranged between 92% and 99%, with an average of 98%. The constructed phylogeny revealed that the population was divided into two major clades. As shown in Fig 3, the BAPS analysis split our dataset further into five groups based on the secondary level of clustering. Groups were monophyletic, confirming the robustness of the major branches of the SNP phylogeny, which were also supported by 100% bootstrap scores.

The SNP analysis demonstrated that the clades comprising the ST-23, ST-27, corresponded to BAPS1 and BAPS2 groups while the ST-24 and ST-25 were included in the BAPS3 and the ST-26 was further subdivided into 2 groups (BAPS4 and BAPS5). The BAPS1 group comprised strains from the *B. ceti* species which contained mostly strains isolated from porpoises. BAPS2 group was composed of two strains from *B. ceti* ST-27, one of which was isolated from a human in New Zealand and the second from a dolphin in Croatia. The BAPS3 was composed of strains from the *B. pinnipedialis* species. The isolates from ST-26 were split into two subpopulations belonging to two different geographical areas with BAPS4 isolated in Costa Rica and BAPS5 isolated in the European Atlantic Sea and in the Mediterranean Sea. Interestingly, BAPS4 and BAPS5 contained mostly isolates from dolphins.

Whole genome SNP analysis showed that the strains from Italy were divided further into two genetically distinct subclades. The first subclade comprised exclusively the isolates from Ionian and Adriatic seas, while the second one included the strains from the Ligurian Sea and

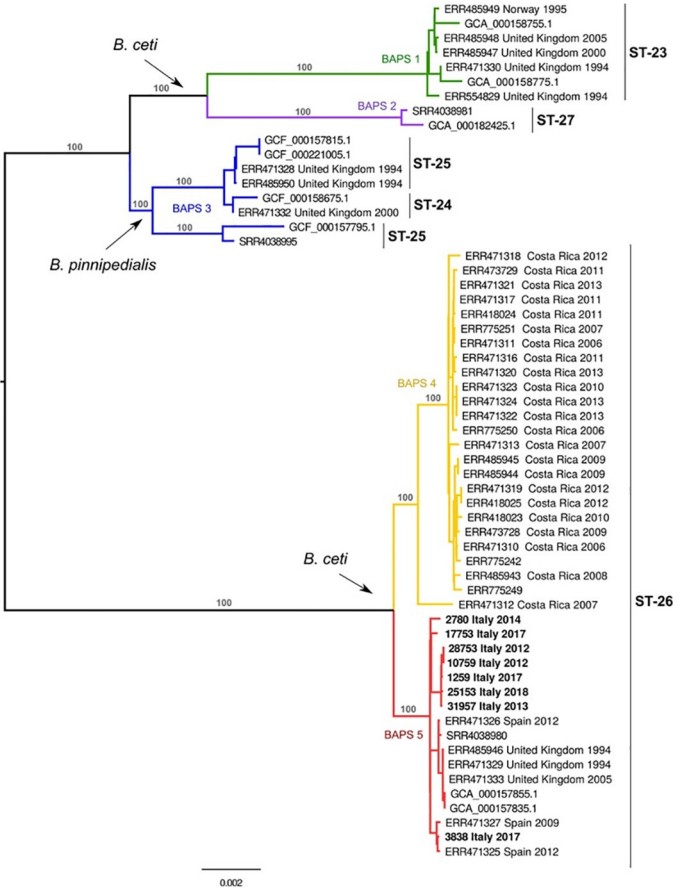

**Fig 3. Maximum likelihood tree of *B. ceti* and *B. pinnipedialis*.** The tree was constructed using concatenated SNP sequences of 59 isolates and mid-point rooted. The branch colours correspond to BAPS populations and the major branches are labelled with bootstrap values.

the Catalonian coast (Fig 3). Plotting these clades onto the geographic map suggests a link between their phylogeny and topographical distribution.

## Discussion

To our knowledge, this study represents the first survey of *B. ceti* infection in striped dolphins from Italian waters and the first extensive characterization of *B. ceti* isolates reported to date.

The isolation of *B. ceti* was achieved from the CNS of all the dolphins under investigation. Only in two cases the isolation was obtained also from the other tissues (specifically spleen for case 1 and spleen and lung for case 5).

Although we observed several not pathognomonic signs, we detected specific gross pathological findings associated with *Brucella* infection, represented by hyperaemia of the meninges [9, 11] in three animals (cases 1, 4, 8).

No lesions were detected in the reproductive organs, and no signs associated with a potential abortion were found in the only adult female sampled (case 6).

A correlation between the infection and the pathological microscopic changes was observed in all cases submitted to histopathological investigations (neurobrucellosis, associated in one case with hepatic and lymph node necrosis). Specifically, non-suppurative meningitis or meningoencephalitis were detected in the CNS of 7 *B. ceti*-infected animals, thereby

recapitulating the features of neurobrucellosis observed in humans [21, 22], as well as in striped dolphins elsewhere [9, 11, 19, 23–26].

We detected severe coinfections in half of the animals investigated (cases 5, 6, 7, 8), involving DMV in all cases and *T. gondii* in case 5, as reported before for several cetacean species infected by *Brucella* spp. and *Cetacean Morbillivirus* [63] or *T. gondii* [26].

Considering the role of *B. ceti* infection in the striped dolphins under study, in cases 1, 2 and 4 the stranding could have resulted from a severe cerebral impairment, associated with severe brain inflammation caused by *B. ceti* infection.

Moreover, in case 2, the finding of a generalized lymphoid depletion, described before in dolphins with brucellosis [18, 64], suggests an immunocompromised host response, though the evidence of hepatocyte vacuolar degeneration could additionally make the effects of toxic environmental pollutants and/or an undisclosed viral infection other than DMV plausible.

The cause of death could not be hypothesized for case 3, considering the limited data available.

In cases 5, 6, 7 and 8 the stranding could have resulted from a severe cerebral impairment, associated with a coinfection by *B. ceti* and DMV, and, for case 5, also *T. gondii*. Noteworthy, in case 5, a striped dolphin in a poor body condition, the systemic spread of *B. ceti* infection, the evidence of additional *Brucella*-type lesions (multicentric lymphoid necrosis and multifocal necrotizing hepatitis), with the absence of anti-*Brucella* spp. antibodies and negative bacteriological and biomolecular results in CSF, suggests an acute fatal brucellosis infection that appears to be the most likely contributing cause of death. DMV and *T. gondii* infections, associated with typical pathological findings, represented respectively by bronchointerstitial pneumonia and protozoan cysts at cerebral level, along with specific immunopositivity and the presence of antibodies for both agents, may indeed support the potentially relevant role played by DMV and *T. gondii* in initiating the animal's decline.

Interestingly, the anti-*Brucella* spp. antibodies observed in case 8 represent the only positive result among the three tested animals. The limited number of samples hampers any conclusion on the use of RBT for the detection of *Brucella* infection in dolphins. Nevertheless, the fact that the positive result was obtained by testing a fresh serum sample collected from an animal in a good conservation code, supports a true positive result [11]. Moreover, the negative serological findings are supported by the simultaneous detection of other antibodies in case 5 and a supposed immunocompromised host response in case 6, in presence of a widespread DMV infection. RBT test, while generally considered consistent, may produce false results due to variety of factors [65]. Therefore, in order to screen the immunological status of the examined animals, other serological tests such as ELISA could be used, as previously shown [66]. Although some discrepancies between results of RBT and ELISA tests have been reported, ELISA tests, such as iELISA have been successfully used to detect anti-Brucella antibodies in odontocetes and arctic wildlife [65, 66] and could therefore serve as a complementary method serological response to *B. ceti* in dolphins. The highest frequency of *B. ceti* infection was confirmed in juveniles. This observation seems to be in discordance with a study performed in striped dolphins stranded in Costa Rica [64], in which meningoencephalomyelitis was revealed in the same number of juveniles and adults that displayed neurological syndromes before death, as well as with a previous study in marine mammals stranded in Brazil [18]. Specifically, in a population made up of several cetacean species, the highest frequency of *Brucella* spp. infection was confirmed in the newborn calves, whereas, within the genus *Stenella*, the most commonly infected age-group were the adults.

None of the cases considered in this report stranded alive, so it was not possible to observe neurological symptoms at the time of stranding.

The genomic analysis of the *B. ceti* strains isolated from the stranded dolphins grouped them into the common ST-26 in agreement with the previous reports [28]. The whole genome SNP analysis showed that the population was divided into 5 main groups, one of which included the *B. pinnipedialis* species while the remaining four were assigned to *B. ceti*. It is interesting to note how these groups identify specific STs in accordance with the MLST which therefore allows us to use previous data as a legacy to give consistency to the host spectrum as well as to geographic location. According to these data, we observed that the BAPS1 from *B. ceti* and BAPS3 from *B. pinnipedialis*, although not exclusively, are mainly found in porpoises and seals. Similar cluster partition was observed previously with MLVA data, which demonstrated the existence of the *B. ceti* group, predominantly associated with porpoises [28]. Their geographical location, instead, appeared to be linked to the Atlantic Ocean, even though the real distribution could be wider.

On the other hand, the BAPS 2, 4 and 5 populations seemed to be mainly observed in dolphins. The more consistent groups, BAPS4 and BAPS5, which belonged to the ST-26, were further divided from a geographical point of view and split the of Costa Ricans strains from those of the Mediterranean and the Atlantic Ocean. Further analysis revealed that the BAPS5 population was divided into geographically separated subgroups. The phylogeographic division found for these strains resembled the phylogeographic evolution of the striped dolphin [67]. The evolution of the striped dolphin suggests that its population is divided between the eastern and western Mediterranean, thus confirming the genomic division between *B. ceti* strains from the Ligurian Sea and the Adriatic and Ionian Seas observed in our analysis.

The data obtained by whole genome SNP analysis suggest an interesting relationship between phylogeny and geographical distribution of *B. ceti* strains in Italy, therefore providing new insights into the phylogenetic structure of *B. ceti* in the Mediterranean. Nevertheless, further studies from the Mediterranean Sea are required to elucidate the molecular evolution of *B. ceti* and its actual distribution.

In summary, our results provide novel data and pathological evidence of *B. ceti* infection in cetacean species in Italy, and the geographic distribution range of this agent in Italian waters. Considering the results of this survey and the other data available [16, 26], the occurrence of *B. ceti* infection in cetaceans stranded, along the Italian coastline appears to be limited to specific areas (Liguria, Tuscany, Apulia, Calabria), with the highest occurrence of *B. ceti* infected cetaceans along the Ionian coastline, which suggests consistent circulation of the bacterial pathogen in that area. Our results highlight the need for continuous surveillance and monitoring studies to better understand the pathogen, host and environmental factors involved in cetacean *Brucella* spp. infection's epidemiology, in tight agreement with the "One Health" concept. WGS typing proved to be useful for molecular classification of *B. ceti* strains and allowed the typing of large populations. The genetic clustering based on SNP analysis was in agreement with all previously reported methods and additionally it provided a much higher discriminatory power.

The severity of *B. ceti*-associated lesions reported in the present study supports the role of this microbial agent as a primary neurotropic pathogen in striped dolphins, as well as a probable cause of stranding events and death, as previously described elsewhere [14]. In this regard, our results corroborate previous reports indicating striped dolphins as highly susceptible hosts for developing neurobrucellosis in comparison with the other cetaceans [64], thus confirming neurobrucellosis as one of the most significant lesions' pattern associated to *B. ceti* infection [26, 64, 68].

Additional studies are required to identify the mechanisms involved in the crossing of the blood-brain barrier and the pathogen and host-related factors driving *B. ceti* neuroinvasion, colonization and persistence in the CNS [27]. Moreover, a detailed understanding of the effects

of pollutant-related immunotoxicity on pathogenicity of *B. ceti*, as suggested by some case reports [16, 69], is required, particularly in light of conflicting result obtained using ex vivo model [70].

Surveillance of strandings in Italy involves organisations from governmental and academic institutions with different areas of expertise such as public health, animal health and environment. Such network made our study possible, and our findings highlight the importance of the multidisciplinary approach in the monitoring of stranded cetaceans, with epidemiological data and laboratory information truly shared across sectors in the perspective of the one-health approach.

Finally, based on the demonstrated zoonotic capability of *B. ceti* [31, 32, 33], proper handling of stranded animals, together with an *ad hoc* adoption of all necessary biosafety and biosecurity measures and protocols during post mortem and diagnostic investigations on stranded cetaceans are strongly recommended to avoid the risk of transmission to humans of this and other zoonotic pathogens.

## Supporting information

**S1 Table. Full description of gross and microscopical findings, associated with complete diagnostic test results and the most likely cause of death for each case considered.**
(DOCX)

## Author Contributions

**Conceptualization:** Giuliano Garofolo, Cristina Casalone, Carla Grattarola.

**Data curation:** Giuliano Garofolo, Antonio Petrella, Giuseppe Lucifora, Gabriella Di Francesco, Giovanni Di Guardo, Barbara Iulini, Carla Grattarola.

**Formal analysis:** Giovanni Di Guardo.

**Funding acquisition:** Walter Mignone.

**Investigation:** Giuliano Garofolo, Antonio Petrella, Giuseppe Lucifora, Alessandra Pautasso, Barbara Iulini, Katia Varello, Federica Giorda, Maria Goria, Alessandro Dondo, Simona Zoppi, Cristina Esmeralda Di Francesco, Stefania Giglio, Furio Ferringo, Luigina Serrecchia, Mattia Anna Rita Ferrantino, Katiuscia Zilli, Anna Janowicz, Walter Mignone, Carla Grattarola.

**Methodology:** Giuliano Garofolo, Antonio Petrella, Simona Zoppi, Manuela Tittarelli.

**Software:** Giuliano Garofolo, Anna Janowicz.

**Supervision:** Carla Grattarola.

**Writing – original draft:** Giuliano Garofolo, Carla Grattarola.

**Writing – review & editing:** Giuliano Garofolo, Antonio Petrella, Giuseppe Lucifora, Gabriella Di Francesco, Giovanni Di Guardo, Alessandra Pautasso, Barbara Iulini, Katia Varello, Federica Giorda, Maria Goria, Alessandro Dondo, Simona Zoppi, Cristina Esmeralda Di Francesco, Stefania Giglio, Furio Ferringo, Luigina Serrecchia, Mattia Anna Rita Ferrantino, Katiuscia Zilli, Anna Janowicz, Manuela Tittarelli, Walter Mignone, Cristina Casalone, Carla Grattarola.

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
