## [Decision Letter · Decision Letter 0]

21 May 2020

PONE-D-20-10982

Occurrence of *Brucella ceti* in striped dolphins from Italian Seas

PLOS ONE

Dear Dr. Grattarola,

Thank you for submitting your manuscript to *PLOS ONE*. Both reviewers expressed the opinion that the studies described in your paper were well done and that the information presented will be valuable to the field. However, both reviewers also pointed out issues that need to be addressed before the manuscript will be considered suitable for publication. Thus, I am going to ask that you submit a revised manuscript that appropriately addresses all of  the issues raised by both of  these reviewers.

We would appreciate receiving your revised manuscript by August 20, 2020. To enhance the reproducibility of your results, we recommend that if applicable you deposit your laboratory protocols in protocols.io, where a protocol can be assigned its own identifier (DOI) such that it can be cited independently in the future. For instructions see: http://journals.plos.org/plosone/s/submission-guidelines#loc-laboratory-protocols

We look forward to receiving your revised manuscript!

Sincerely,

R. Martin Roop II, Ph.D.

Academic Editor

PLOS ONE

Journal Requirements:

2. We note that you are reporting an analysis of a microarray, next-generation sequencing, or deep sequencing data set. PLOS requires that authors comply with field-specific standards for preparation, recording, and deposition of data in repositories appropriate to their field. Please upload these data to a stable, public repository (such as ArrayExpress, Gene Expression Omnibus (GEO), DNA Data Bank of Japan (DDBJ), NCBI GenBank, NCBI Sequence Read Archive, or EMBL Nucleotide Sequence Database (ENA)). In your revised cover letter, please provide the relevant accession numbers that may be used to access these data. For a full list of recommended repositories, see http://journals.plos.org/plosone/s/data-availability#loc-omics or http://journals.plos.org/plosone/s/data-availability#loc-sequencing.

3. In your Methods section, please provide additional location information of the study sites, including geographic coordinates for the data set if available.

Reviewers' comments:

Reviewer's Responses to Questions

**Comments to the Author**

1. Is the manuscript technically sound, and do the data support the conclusions?

Reviewer #1: Yes

Reviewer #2: Yes

2. Has the statistical analysis been performed appropriately and rigorously? 

Reviewer #1: Yes

Reviewer #2: N/A

3. Have the authors made all data underlying the findings in their manuscript fully available?

Reviewer #1: Yes

Reviewer #2: Yes

4. Is the manuscript presented in an intelligible fashion and written in standard English?

Reviewer #1: Yes

Reviewer #2: No

5. Review Comments to the Author

Reviewer #1: General comments

This is a very comprehensive study. However, some clarification is needed (see specific comments).

I would have like to see some comparison with MLVA results described in the manuscript. The PCR methods for identification could have been using another target (IS711) and/or be updated (qPCR).

About pathology, the authors should discuss abortion (they seem not to have seen).

The authors should mention that they did not sample in the Adriatic Sea. They should discuss that this may be one of the reasons why they did not report ST27.

Specific comments

L61: lesions are described in cetaceans not in pinnipeds, so change “marine mammals” accordingly.

L91: the first report on the isolation of B. ceti was in 1994. Do you mean first isolation in the Mediterranean sea?

L163: why were subcultures performed on Farrell’s media?

L196: outer membrane of Brucella spp., not B. abortus. Why did you not use the IS711 based PCR or even better qPCR?

L260: BAPS analysis (ref 57, 58) are not been used previously for Brucella. A short explanation is needed here.

L268-432: I am wondering if all the case descriptions are necessary. Perhaps Table 1, would be sufficient?

L463: which “other genomes”?

L537-538: please explain how these results are in agreement with MLVA results.

L556-557: please be more specific. An association between exposure to pollutants and higher exposure/infection rates in marine mammals has been suggested (you touch upon these things later in the manuscript).

L578: could you please explain to the prospective reader your multidisciplinary approach?

Reviewer #2: The manuscript by Garofolo and coworkers entitled “Occurrence of Brucella ceti in striped dolphins from Italian Seas” describes the pathological analysis of 8 cases of stranding of striped dolphins in the coasts of Italy. Several diagnostic tools are used in order to investigate cause of stranding, including macro and microscopic observations, histology, serology, molecular detection, bacterial isolation, parasites identification in target tissues according to literature. In the presented cases, bacterial cultures were positive for Brucella and results confirmed by WGS.

The study is methodological sound and the analysis is thorough.

The way data is presented could be improved, particularly to improve focus as follows:

1. Normalize the detail level of findings described. Although emphasis is given in Brucella ceti findings and characterization, there are other interesting findings related to the stranded dolphins that deserve attention. This should also be reflected in the materials and methods section, where description of protocols used to carry out the pathology study, parasites and worm identification as well as histological findings are need it.

2. Remove description of cases, and add it as a supplementary info. Description of each case can be improved connecting ideas instead of listing findings. Use table 1 to improve results section writing.

3. In the results section, indicate genome quality of the genomes analyzed, including depth coverage and percentage of mapping against reference genome. Were these criteria used in the selection of genomes to be included in the analysis.

4. Several possible cause of deaths are suggested for the different cases. Indicate how the authors arrived to the different conclusions indicated for each case.

5. Explain how the IonTorrent sequenced genomes were used in the presented work.

Minor points:

1. Introduction, page 10, lane 80: “The marine Brucella strains are classified into three major groups, with five clusters and 15 sequence 81 types (STs). Please explain what are the major groups and clusters and the context of this classification.

2. Materials and methods, page 13, lane 152: “ The CNS of the case 6 was submitted to isolation retrospectively, since histopathology findings of neurobrucellosis had been observed”. Not sure what is the message from this sentence; retrospectively means that was carried out after a long time has passed since the histopathology analysis?

3. Materials and methods, page 13, lane 162: " The enrichment cultures in broth media were weekly subcultured (six subcultures) on Farrell’s Agar medium. Suspect colonies were tested for Gram stain reaction as well as for catalase, oxidase and urease activities, motility and slide agglutination tests with Brucella polyvalent antiserum.” Differences in smooth and rough phenotype have being described after several passages in B. ceti isolates. Please indicate on what subculture the agglutination test was performed and specifically what Brucella polyvalent antiserum was used.

4. Materials and methods, page 15, lane 203: please specify the negative controls used. This is very important since the used PCR was not designed to detect Brucella specific amplicons in striped dolphins’ tissues.

5. Discussion, page 27, first lane: “To our knowledge, this study represents the first survey of B. ceti infection in cetaceans of Italian waters and the first extensive characterization of B. ceti isolates reported to date”. Cetaceans is a general group of marine mammals. The study refers to a deep analysis of B. ceti isolates from striped dolphins.

6. Discussion, page 27, lane 496 : “Considering the major gross and microscopical findings reported, many of the general pathological findings were not related to brucellosis.” This affirmation seems contradictory to the explanation that follows it. Please rephrase.

7. Discussion, page 28, lane 514: “Based on serological investigations, only one case (Case 8) of the three investigated (Cases 5, 6, 8) resulted positive for anti-Brucella spp. antibodies.” This observation deserves further discussion, since coinfections are relevant in the Mediterranean. Why only one positive for RBT? Causes? Sample degradation? Bacteria with rough phenotype? How is this related with the fact that the authors state that Brucella appears to be acting as a secondary pathogen in lane 519? Brucella has been considered a primary pathogen in most host species. If the authors consider that this is not the case in striped dolphins, this should be properly discussed.

8. Discussion, page 28, lane 521 and following: “The highest frequency of B. ceti infection was confirmed in juveniles (6 out of 7 cases with age determined), followed by adults (1 out of 7).” Comparison with the data from Brazil are not accurate since sample population is different, taking into account several cetacean species. A more accurate comparison can be done by comparing only striped dolphin’s data in the Brazil study, and recent Costa Rica and US studies.

6. PLOS authors have the option to publish the peer review history of their article (what does this mean?). If published, this will include your full peer review and any attached files.

Reviewer #1: Yes: Jacques Godfroid

Reviewer #2: Yes: Caterina Guzmán-Verri

---

## [Author Response · Author response to Decision Letter 0]

11 Aug 2020

Dear Reviewer 1, 

We would like to thank you for giving us the opportunity to revise and resubmit this manuscript.

We answered to all general and specific comments, point by point, in the Response to Reviewers in attachment.

Overall, we have extensively edited the manuscript in order to avoid the descriptions of all the cases, 

by a reorganization of the material methods and results sections, also creating a supplementary table, to normalize the detail level of findings in the text.

We hope that you will find that we have addressed all the concerns raised.

Dear Reviewer 2,

We would like to thank you for giving us the opportunity to revise and resubmit this manuscript.

We answered to all comments, point by point, in the Response to Reviewers in attachment.

Specifically, in order to normalize the detail level of findings described, we have extensively edited the manuscript avoiding the descriptions of all the cases,and we reorganized the material methods and results sections, also creating a supplementary table to give more details on the ancillary tests. Moreover, we revised in the conclusions about the possible cause of death suggested for the differents cases under study, and you may check it in the discussion section.

We have had one English speaker review the manuscript that has found this current version to be grammatically sound. I trust that the editing improved the manuscript.

We hope that you will find that we have addressed all the concerns raised.

---

## [Decision Letter · Decision Letter 1]

10 Sep 2020

PONE-D-20-10982R1

Occurrence of *Brucella ceti* in striped dolphins from Italian Seas

PLOS ONE

Dear Dr. Grattarola,

Both reviewers expressed the opinion that your revised manuscript is greatly improved, but they both also noted a few things that still need to be addressed. Their comments should be easy to address, and doing so will further improve the paper. Thus, I am going to ask that you submit a revised manuscript that addresses the points they raise.

Please submit your revised manuscript by December 9, 2020. However, I don't think that it will take very long to deal with the few points the reviewers have raised. But if  you will need more time than this to complete your revisions, please reply to this message or contact the journal office at plosone@plos.org. Please include the following items when submitting your revised manuscript:

I look forward to receiving the revised manuscript!

Sincerely,

Marty Roop

Academic Editor

PLOS ONE

Reviewers' comments:

Reviewer's Responses to Questions

**Comments to the Author**

1. If the authors have adequately addressed your comments raised in a previous round of review and you feel that this manuscript is now acceptable for publication, you may indicate that here to bypass the “Comments to the Author” section, enter your conflict of interest statement in the “Confidential to Editor” section, and submit your "Accept" recommendation.

Reviewer #1: All comments have been addressed

Reviewer #2: (No Response)

2. Is the manuscript technically sound, and do the data support the conclusions?

Reviewer #1: Yes

Reviewer #2: Yes

3. Has the statistical analysis been performed appropriately and rigorously? 

Reviewer #1: Yes

Reviewer #2: N/A

4. Have the authors made all data underlying the findings in their manuscript fully available?

Reviewer #1: Yes

Reviewer #2: Yes

5. Is the manuscript presented in an intelligible fashion and written in standard English?

Reviewer #1: Yes

Reviewer #2: Yes

6. Review Comments to the Author

Reviewer #1: I have 1 comment that perhaps need an explanatory sentence and reference in the final manuscript.

Serology:

The RBT has limitations when used in wildlife species, particularly marine mammals. therefore ELISAs are excellent alternatives. More, a "universal" indirect ELISA for detecting antibodies in mammals, including marine mammals has been described and documented: Nymo I.H., Godfroid J., Asbakk K., Larsen A.K., das Neves C.G., Rødven R., Tryland M. 2013. A protein A/G indirect enzyme-linked immunosorbent assay for the detection of anti-Brucella antibodies in Arctic wildlife. J. Vet. Diagn. Invest., 25: 369-375. doi: 10.1177/1040638713485073.

I feel that reference to ELISA is needed. I trust the RBT results reported in this study. However, "true" result may be misleading. Indeed, infected animals may be classified negaltive by RBT but positive by iELISA. So, a word of explanation is needed here.

Additional comment about pollution and immunocompromition:

Actually, the link is difficult to demonstrate beyond a general statement.

Nymo I.H., das Neves C.G., Tryland M., Bårdsen B.J., Santos R.L., Turchetti A.P., Janczak A.M., Djønne B., Lie E., Berg V., Godfroid J. 2014. Brucella pinnipedialis hooded seal (Cystophora cristata) strain in the mouse model with concurrent exposure to PCB 153. Comp. Immunol. Microbiol. Infect. Dis., 37: 195-204. doi: 10.1016/j.cimid.2014.01.005.

So, a word of caution could be needed.

Reviewer #2: This version of the manuscript is a good improvement from the previous version. There are some issues that I think are important to address, related mainly with the description of methods used. This can help to reproduce data by independent researchers.

Materials and Methods section

Move:

252All strains isolated from the striped dolphins under study were identified asassigned to B. ceti using

253 the PCR-RFLP method [51149] and then subjected to genomic analysis at the OIE and National and

254 OIE Reference Laboratory for Brucellosis, Istituto Zooprofilattico Sperimentale dell’of Abruzzo e

255 del and Molise, Teramo, Italy.

To the WGS section and rephrase accordingly, assuming that the same DNA extraction method was used.

261 The reactions were loaded as previously reported using B. suis bv 2 strain Thomsen and no template

262 control as positive and negative controls, respectively

No clear to this reviewer what this means. Please specify what was added in a no template control as positive and what is a negative control under this context. A proper negative control should include a sample from a similar animal or tissue from the same animal where no isolation of Brucella was obtained. No evidence of standardization of this PCR for cetacean tissues is given, therefore, this control is essential

As stated by the authors in their response, please include the following info the related to selection criteria of the public genomes used in the study:

“We did however limit our dataset to non-identical sequences and the sequences that mapped to the reference with less than 500 ambiguous matches.”

According to the answer given by the authors:

"As described in Materials and Methods section, two strains of B. ceti isolated in Italy (10759 and 28753) were sequenced by our laboratory previously using Ion Torrent technology."

And a quick database search, it seems that these sequences were already published. If that is the case, please refer properly.

Results section

First lane: “Post-mortem and histopathological investigations were performed on seven positive animals”

I assume it means seven of the eight animals with positive culture for Brucella?

Supplementary Table 2

Please add references for all methods used, particularly those used for parasites

7. PLOS authors have the option to publish the peer review history of their article (what does this mean?). If published, this will include your full peer review and any attached files.

Reviewer #1: **Yes: **Jacques Godfroid

Reviewer #2: No

---

## [Author Response · Author response to Decision Letter 1]

18 Sep 2020

Reviewer #1:

Reviewer #1: I have 1 comment that perhaps need an explanatory sentence and reference in the final manuscript.

Serology:

The RBT has limitations when used in wildlife species, particularly marine mammals. therefore ELISAs are excellent alternatives. More, a "universal" indirect ELISA for detecting antibodies in mammals, including marine mammals has been described and documented: Nymo I.H., Godfroid J., Asbakk K., Larsen A.K., das Neves C.G., Rødven R., Tryland M. 2013. A protein A/G indirect enzyme-linked immunosorbent assay for the detection of anti-Brucella antibodies in Arctic wildlife. J. Vet. Diagn. Invest., 25: 369-375. doi: 10.1177/1040638713485073.

I feel that reference to ELISA is needed. I trust the RBT results reported in this study. However, "true" result may be misleading. Indeed, infected animals may be classified negaltive by RBT but positive by iELISA. So, a word of explanation is needed here.

We thank the reviewer for the suggestion, we modified the text accordingly. Now it reads:

”The limited number of samples hampers any conclusion on the use of RBT for the detection of Brucella infection in dolphins. Nevertheless, the fact that the positive result was obtained by testing a fresh serum sample collected from an animal in a good conservation code, supports a true positive result [11]. Moreover, the negative serological findings are supported by the simultaneous detection of other antibodies in case 5 and a supposed immunocompromised host response in case 6, in presence of a widespread DMV infection. RBT test, while generally considered consistent, may produce false results due to variety of factors [65]. Therefore, in order to screen the immunological status of the examined animals, other serological tests such as ELISA could be used, as previously shown [66]. Although some discrepancies between results of RBT and ELISA tests have been reported, ELISA tests, such as iELISA have been successfully used to detect anti-Brucella antibodies in odontocetes and arctic wildlife [65, 66] and could therefore serve as a complementary method serological response to B. ceti in dolphins.”

Additional comment about pollution and immunocompromition:

Actually, the link is difficult to demonstrate beyond a general statement.

Nymo I.H., das Neves C.G., Tryland M., Bårdsen B.J., Santos R.L., Turchetti A.P., Janczak A.M., Djønne B., Lie E., Berg V., Godfroid J. 2014. Brucella pinnipedialis hooded seal (Cystophora cristata) strain in the mouse model with concurrent exposure to PCB 153. Comp. Immunol. Microbiol. Infect. Dis., 37: 195-204. doi: 10.1016/j.cimid.2014.01.005.

So, a word of caution could be needed.

We thank the reviewer for the observation. We wrote again the sentence to make the statement more appropriate. Now it reads: 

Moreover, a detailed understanding of the effects of pollutant-related immunotoxicity on pathogenicity of B. ceti, as suggested by some case reports (16, 69), is required, particularly in light of conflicting result obtained using ex vivo model (70).

Line 470.

Reviewer #2:

Reviewer #2: This version of the manuscript is a good improvement from the previous version. There are some issues that I think are important to address, related mainly with the description of methods used. This can help to reproduce data by independent researchers.

Materials and Methods section

Move:

252All strains isolated from the striped dolphins under study were identified asassigned to B. ceti using

253 the PCR-RFLP method [51149] and then subjected to genomic analysis at the OIE and National and

254 OIE Reference Laboratory for Brucellosis, Istituto Zooprofilattico Sperimentale dell’of Abruzzo e

255 del and Molise, Teramo, Italy.

To the WGS section and rephrase accordingly, assuming that the same DNA extraction method was used.

Revised accordingly. The DNA extraction was moved at the line 204.

261 The reactions were loaded as previously reported using B. suis bv 2 strain Thomsen and no template

262 control as positive and negative controls, respectively

No clear to this reviewer what this means. Please specify what was added in a no template control as positive and what is a negative control under this context. A proper negative control should include a sample from a similar animal or tissue from the same animal where no isolation of Brucella was obtained. No evidence of standardization of this PCR for cetacean tissues is given, therefore, this control is essential.

We regret that the sentence was unclear but in the statement we reported the use of B. suis bv 2 strain Thomsen as positive control and no template control for the negative control. Typically, a negative control for PCR is one sample that should not give amplicons, therefore, any visible bands might be a result of contamination or non-specific amplification. The lab works under ISO/IEC 17025 regulation and has validation procedure for the diagnostic tests therefore the PCR was analytically validated for detecting Brucella DNA.

Moreover using tissue samples from bacteriological negative animals could be misleading because there is the possibility of finding dead cells or free DNA from Brucella that gives you positive PCR reactions and negative bacteriological isolation. This situation could be even more evident using tissue from the same animal where you may find positive bacteriological results from one tissue and negative bacteriological from another one. 

We revised for clarity the line 219. Now it reads: “The reactions were loaded as previously reported using B. suis bv 2 strain Thomsen as positive control and no template control as negative control.”

As stated by the authors in their response, please include the following info the related to selection criteria of the public genomes used in the study:

“We did however limit our dataset to non-identical sequences and the sequences that mapped to the reference with less than 500 ambiguous matches.”

As suggested, we inserted the following sentence in Materials and methods section: 

“The dataset was limited to non-identical sequences that mapped to the B. ceti reference genome with less than 500 ambiguous matches (GenBank Accession Numbers NC_022905.1; NC_022906.1)”

Line 259. 

According to the answer given by the authors:

"As described in Materials and Methods section, two strains of B. ceti isolated in Italy (10759 and 28753) were sequenced by our laboratory previously using Ion Torrent technology."

And a quick database search, it seems that these sequences were already published. If that is the case, please refer properly.

Revised accordingly, citing the reference 42: 

Ancora M, Marcacci M, Orsini M, Zilli K, Di Giannatale E, Garofolo G, et al. Complete Genome Sequence of a Brucella ceti ST26 Strain Isolated from a Striped Dolphin (Stenella coeruleoalba) on the Coast of Italy. Genome Announc. 2014 Mar 6;2(2). pii: e00068-14. doi: 10.1128/genomeA.00068-14. 

Line 257.

Results section

First lane: “Post-mortem and histopathological investigations were performed on seven positive animals”

I assume it means seven of the eight animals with positive culture for Brucella?

Yes. We revised for clarity the line, specifying appropriately.

Line 278.

Supplementary Table 2

Please add references for all methods used, particularly those used for parasites

We thank the reviewer for the suggestion. We modified the Supplementary Table 2 accordingly, and we added the references for methods used for parasites in the Section “Necropsy and diagnostic investigations”:

Line 161.

---

## [Editor Report · Decision Letter 2]

22 Sep 2020

Occurrence of *Brucella ceti* in striped dolphins from Italian Seas

PONE-D-20-10982R2

Dear Dr. Grattarola,

Thank you for your quick turnaround on the manuscript! I'm pleased to inform you that it has now been judged scientifically suitable for publication and will be formally accepted once it meets any necessary technical requirements.

Sincerely,

R. Martin Roop II, Ph.D.

Academic Editor

PLOS ONE
---

## [Editor Report · Acceptance letter]

25 Sep 2020

PONE-D-20-10982R2 

Occurrence of *Brucella ceti* in striped dolphins from Italian Seas 

Dear Dr. Grattarola:

I'm pleased to inform you that your manuscript has been deemed suitable for publication in PLOS ONE. Congratulations! Your manuscript is now with our production department. 

Kind regards, 

on behalf of

Dr. Roy Martin Roop II 

Academic Editor

PLOS ONE